# Can SARS-CoV-2 Infection Lead to Neurodegeneration and Parkinson’s Disease?

**DOI:** 10.3390/brainsci11121654

**Published:** 2021-12-18

**Authors:** Lea Krey, Meret Koroni Huber, Günter U. Höglinger, Florian Wegner

**Affiliations:** Department of Neurology, Hannover Medical School, Carl-Neuberg-Straße 1, 30625 Hannover, Germany; Huber.Meret@mh-hannover.de (M.K.H.); Hoeglinger.Guenter@mh-hannover.de (G.U.H.); wegner.florian@mh-hannover.de (F.W.)

**Keywords:** Parkinson’s disease, COVID-19, SARS-CoV-2, neurodegeneration, Alzheimer’s disease, viral infection

## Abstract

The SARS-CoV-2 pandemic has affected the daily life of the worldwide population since 2020. Links between the newly discovered viral infection and the pathogenesis of neurodegenerative diseases have been investigated in different studies. This review aims to summarize the literature concerning COVID-19 and Parkinson’s disease (PD) to give an overview on the interface between viral infection and neurodegeneration with regard to this current topic. We will highlight SARS-CoV-2 neurotropism, neuropathology and the suspected pathophysiological links between the infection and neurodegeneration as well as the psychosocial impact of the pandemic on patients with PD. Some evidence discussed in this review suggests that the SARS-CoV-2 pandemic might be followed by a higher incidence of neurodegenerative diseases in the future. However, the data generated so far are not sufficient to confirm that COVID-19 can trigger or accelerate neurodegenerative diseases.

## 1. Introduction

The aim of this review is to summarize the data on the link between COVID-19, other viral infections and Parkinson’s disease. Therefore, different mechanisms of the SARS-CoV-2 virus affecting cerebral functions and inducing neurodegeneration have to be considered. First, a direct neurotoxic effect of the virus resulting from neuroinvasion is possible as well as secondary effects due to systemic inflammatory alterations. In the first section, we present the knowledge on SARS-CoV-2 neurotropism, neuropathology, neuroinflammation and changes in levels of biomarkers that have been observed during infection. This is followed by a brief overview on the connection between other viral infections and neurodegenerative diseases. In the third chapter, we highlight the link between COVID-19 and neurodegeneration focusing on Parkinson’s disease (PD) and cognitive dysfunction/Alzheimer’s disease (AD). At last, we discuss how the pandemic has influenced symptoms and psychosocial aspects in PD patients underlining the tremendous impact that the infection has had especially on people with preexisting neurological conditions and disabilities.

## 2. Methods

Literature research for this review was done in PubMed using the search terms “COVID-19”, “SARS-CoV-2”, “Parkinson’s disease”, “Alzheimer’s disease”, “neurodegeneration”, “viral infection” and “infection” in different combinations. Only articles published in English in international peer-reviewed journals were included in the selection process. Articles were selected by screening the abstracts for eligibility; publications contributing relevant data to the core content of this review were included.

## 3. Chapter 1

### 3.1. SARS-CoV-2 Neurotropism

SARS-CoV-2, the cause of the current pandemic, belongs to the well-known family of Coronaviridae. Previously defined Coronaviridae (HCov-OC43, HCov-229E, SARS-CoV, MERS-Cov) were detected in human brain samples, which proves their neurotropism and their ability to cause persistent infections of the central nervous system (CNS) [1,2,3]. As early as 1999 it was shown in vitro that neuroblastoma, neuroglioma and glial cells were susceptible to infection with human Coronaviridae and that the infection could persist for at least 130 days of culture time [2]. In animal models, this persistent infection led to neuronal loss and long-term sequelae such as reduced activity and hippocampal neuronal volumes as signs of a neurodegenerative phenotype [4,5,6].

There is an ongoing debate about whether SARS-CoV-2 can enter and persist in cerebral structures. Indeed, there are some data supporting the theory of SARS-CoV-2 neurotropism that will be outlined in the following paragraphs.

The ACE-2-receptor was identified as one of SARS-CoV-2′s most common entry receptors [7]. However, it is not highly expressed in the brain compared to other tissues [7]. Its expression in the CNS was shown on glial cells (astrocytes), capillary endothelium, monocytes/macrophages and neurons [8,9]. A comparably high expression of ACE-2-receptor was detected in brainstem areas leading to the hypothesis that SARS-CoV-2 invasion might impair brainstem structures involved in regulating cardiovascular functions [10]. An infection of ACE-2-receptor transgenic mice also led to the expression of viral antigens in neurons especially in the thalamus, cerebrum and brainstem, while the cerebellum remained uninfected [5]. The affected brain areas showed neuronal loss and microglial activation in the absence of other inflammatory signs [5].

In contrast, there is increasing evidence that the ACE-2-receptor may not be the primary way of invasion into the CNS. Instead, other receptors such as neuropilin 1 (NRP-1) could contribute substantially to the invasion of SARS-CoV-2 into cerebral structures [11,12,13]. NRP-1 was found to be highly expressed in neurons and astrocytes [12].

There are three main routes that are speculated to lead to SARS-CoV-2 invasion into the CNS that are presented below [10,11,14,15,16,17,18].

The first possible way of viral invasion is the transneural way starting in the nasal epithelium and the olfactory nerves progressing into the brain via axonal transport [5,14,19]. This route of neurotropism was shown for SARS-CoV and HCov-OC43 after intranasal infection [4,20]. In transgenic ACE-2-receptor expressing mice, intranasal infection with SARS-CoV resulted in neuronal loss [3,20]. Additionally, the mouse equivalent of the human coronavirus, the mouse hepatitis virus, entered the brain via the olfactory nerve after intranasal inoculation [21].

Supporting this idea of neurotropism is the fact that COVID-19 frequently causes olfactory dysfunction (OD). A wide variation in the incidence of OD associated with COVID-19 (5–98%) was observed mostly due to missing objective testing [22]. An Iranian study using objective smell testing in 60 COVID-19 patients revealed that 98% demonstrated smell loss, but only 35% were subjectively aware of their OD, which underlines the importance of objective testing for this symptom [23,24]. Gustation dysfunctions are also common in COVID-19 and can be confused with olfactory problems [25]. OD appears to be a very early symptom in the course of COVID-19 [26]. In general, two mechanisms can lead to OD: First, an obstruction of the olfactory cleft by swelling or rhinorrhea, which could not be detected in COVID-19 patients [24,25,27]. Secondly, defects of sensorineural transmission can impair the sense of smell [25]. A detailed imaging study using CT and MRI on COVID-19 patients with prolonged OD (minimum 1 month) revealed decreased volumes of the olfactory bulb (43.5%) and shallow olfactory sulci (60.9%) as evidence for this underlying pathology of OD in COVID-19 [27]. However, ACE-2-receptors are absent in olfactory sensory neurons and can only be found in supporting cells such as sustentacular cells and horizontal basal cells (reserve stem cells) in the olfactory and respiratory epithelium [22,28]. It has to be acknowledged that OD is generally a common symptom in the elderly, as it occurs in 10% of people over 65 years and in 62–80% over 80 years [24]. OD is also known to be a common symptom in early PD and AD [24]. Interestingly, OD in COVID-19 occurs more often in younger patients and is inversely correlated with death [29]. This supports the contrasting hypothesis that OD is a sign of defense against the virus to prevent it from reaching cerebral structures rather than allowing entry into the CNS [30].

Alternative transneural ways of CNS invasion by SARS-CoV-2 via the trigeminal or vagal nerve have also been discussed [10,11].

The second proposed route of viral invasion into the CNS is the hematogenous pathway with subsequent crossing of the blood–brain barrier or infection of the choroid plexus [10,11,14,15]. This was described for various other viruses, e.g., HIV, HSV, HCMV and enteroviruses [6].

The endothelial cells in blood vessels and choroid plexus could be the target of invasion in this route of infection, as they were shown to express the ACE-2-receptor [8,11]. Additionally, the SARS-CoV-2 spike protein can cross and impair the blood–brain barrier itself by inducing an inflammatory response within the microvascular endothelium [31,32].

Another mechanism supporting this route of invasion could be an increase of the blood–brain barrier permeability due to elevated IL-6 levels that are present in acute COVID-19 disease [14,33].

The third possible pathway of neurotropism for SARS-CoV-2 is the so-called the “Trojan horse mechanism” which describes the viral infection of immune cells (neutrophils, macrophages, monocytes, CD4^+^-lymphocytes) that reach the CNS via bloodstream and then migrate into cerebral structures by diapedesis [10,11,12,15,16,17,18]. Once they are in the cerebral tissue, the virus or viral particles could be released by those immune cells [12].

### 3.2. SARS-CoV-2 Neuropathology

In general, neuropathological hallmarks of COVID-19 autopsy cases are diffuse edema, gliosis with activation of microglia and astrocytes, ischemic lesions, intracranial bleeds, arteriosclerosis, hypoxic–ischemic injury, encephalitis/meningitis and diffuse inflammation [34,35]. Patients suffering from severe COVID-19 showed a reduction in the numbers of neurons and an elevation in the number of activated microglial cells and astrocytes, as well as higher levels of proinflammatory cytokines measured by qPCR [36].

Matching the hypothesis of hematogenous invasion into the brain, Paniz-Mondolfi et al. detected the virus in capillary endothelium and in neurons in frontal lobe tissue from a patient with COVID-19 [11,37]. The virus was not observed in glial cells in vivo [11]. Another group similarly found SARS-CoV-2 to favor CNS endothelial cells with the ACE-2-receptor expressed in smooth muscle cells of blood vessels [38]. Small vessel disease was identified in five out of nine COVID-19 autopsy cases; however, SARS-CoV-2 was only detected in one case using immunohistochemistry [39]. Detection of SARS-CoV-2 in the brain using PCR was equally difficult; the highest viral load was documented in the olfactory bulb, while SARS-CoV-2 PCR was repeatedly negative in the substantia nigra [30,34,40]. Viral presence is, however, rarely detected in viral encephalitis in general (e.g., in herpesvirus-, arbovirus- or enterovirus-induced encephalitis) [6].

The brains of COVID-19 autopsy cases showed microglial activation in the olfactory bulb, frontal cortex, hippocampus and most prominently in the brainstem, whereas lymphocytes did not appear to be activated [39]. Interestingly, patients with a history of delirium during COVID-19 demonstrated more microglial activation in the hippocampus [39]. Patients with and without sepsis could not be distinguished neuropathologically, contradicting the common hypothesis that neuropathology develops secondary to a cytokine storm during septic disease [39].

### 3.3. SARS-CoV-2 Neuroinflammation/Biomarkers

Apart from a direct influence of SARS-CoV-2 on the brain by invasion into the CNS, secondary effects on cerebral functions due to systemic alterations in the course of the disease are widely discussed. Investigations of brain tissue, biofluids and the systemic reaction showed a (neuro-)inflammatory response triggered by COVID-19.

Multiple cytokines were found to be elevated in the blood during acute COVID-19, while increased levels of proinflammatory markers were not detected in the cerebrospinal fluid (CSF) [41]. Serum levels of IL-4, IL-10, IL-6 and IL1β were elevated in COVID-19 patients [33,42,43]. IL-1- and IL-6 are known to trigger neuroinflammation [9].

SARS-CoV-2 antibodies were frequently detected in the CSF of COVID-19 patients, although this does not prove intrathecal antibody production [41,44,45]. The detection of the virus via PCR from CSF was impossible in most cases [41,44,45]. Only few authors described sporadic positive results in SARS-CoV-2 PCR from CSF in patients with severe cerebral symptoms [46,47,48].

Analysis of markers indicating CNS lesions revealed elevated levels of neurofilament light chain (NfL) and glial fibrillary acidic protein (GFAP) in plasma of patients with moderate to severe COVID-19 [17,49]. Additionally, three of eight patients with severe COVID-19 had signs of a disrupted blood–brain barrier, one had a specific intrathecal antibody synthesis and four were positive for 14-3-3 in the CSF [44]. The data on CSF pleocytosis are controversial so far. A case series of 15 patients and a review summarizing CSF white blood cell counts of 409 COVID-19 patients with neurological symptoms observed frequent pleocytosis (defined as >5 cells/µL) in 36% of 15 and 17% of 409 cases [30,50]. On the other hand, a case series of 13 patients with COVID-19 and encephalopathy or seizures reported CSF pleocytosis in only one case, similar to a study with 18 patients with COVID-19 and neurological complications that discovered pleocytosis in four cases and reported all four to be likely due to blood contamination [51,52].

Sun et al. investigated the cargo of neuronal-enriched extracellular vesicles and interestingly found elevated NfL, amyloid-β, neurogranin, tau and phosphorylated tau in COVID-19 patients implicating possible neurodegenerative processes [42].

Take home messages of Chapter 1 (Section 3):It is likely that SARS-CoV-2 can be neurotropic, since this was shown for other human coronaviruses (HCov-OC43, Hcov-229E, SARS-CoV, MERS-Cov) in the past.There are three possible routes of SARS-CoV-2 neuroinvasion: The transneural route via the olfactory nerve, the hematogenous route via vascular endothelium or a permeable blood–brain barrier and the “Trojan-horse-mechanism” by infiltration of immune cells and subsequent invasion into the CNS via diapedesis.Neuropathologically, SARS-CoV-2 leads to microglial activation in distinct CNS areas.SARS-CoV-2 triggers a neuroinflammatory response with increased serum levels of several cytokines (e.g., IL-1, IL-6) and elevated markers such as NfL and GFAP in the CSF indicating CNS lesions.

## 4. Chapter 2

### Viral Infection and Neurodegeneration

Besides the COVID-19 pandemic, there is broad (epidemiological) evidence linking other viral infections to neurodegenerative diseases, especially PD and AD, which will be reviewed in the following chapters.

The idea, that viral infections can promote neurodegeneration first developed with encephalitis lethargica after the Spanish flu epidemic at the beginning of the 20th century [53]. Since then, a connection between infections and neurodegenerative diseases has been assumed repeatedly.

A meta-analysis of 287,773 PD patients and 7,102,901 controls revealed that individuals with reported infections in the past had an elevated risk for PD (odds ratio, 1.20) [54]. This effect was foremost attributable to bacterial infections [54]. In line with this, a more recent study found a “higher infectious burden” defined by the existence of more antibodies against different viruses and bacteria in the blood of PD patients [55]. More specifically, PD risk was shown to be elevated after VZV infection (adjusted hazard ratio, 1.17) and PD patients were more often seropositive for EBV [56,57]. HCV is a well-established risk factor for PD as is HSV-1 infection for the development of AD [58,59,60,61].

Influenza viruses were brought into context with PD, since encephalitis lethargica had a parkinsonian phenotype and an influenza virus was proposed as the infectious agent of the Spanish flu [53]. Furthermore, H1N1 infection led to persistent microglial activation as a sign of chronic neuroinflammation in wildtype mice [62]. H5N1 accordingly led to microglial activation and α-synuclein aggregation in mice resulting in the loss of dopaminergic neurons in the substantia nigra, which is recognized as the pathological hallmark of PD [63]. Furthermore, the influenza A virus was detected postmortem in the substantia nigra of PD patients [64]. A recent case–control study using data from the Danish National Patient Registry revealed that an influenza diagnosis was associated with the development of PD up to ten years later (odds ratio 1.73) [65]. Thus, a strong association between influenza viruses and PD is suspected but needs to be further elucidated.

Japanese encephalitis virus causes a parkinsonian phenotype during acute disease, but even persistent parkinsonism with MRI lesions in the substantia nigra was observed three to five years after viral infection [66].

West Nile virus can also induce parkinsonism during acute infection. In postmortem studies, elevated α-synuclein levels were detected in patients infected with West Nile virus [57,67,68]. An interesting hypothesis about the function of α-synuclein was developed in an α-synuclein-knockout mouse model after West Nile infection [67]. The absence of α-synuclein in this model led to disastrous disease progression, suggesting a protective role of α-synuclein against viral infection [57,67]. It was postulated that α-synuclein entraps viral particles as a cellular defense mechanism, which persists after the infection leading to its pathological aggregation and subsequent neurotoxic effects. The same mechanism was proposed for β-amyloid, which can entrap HSV-1 and inhibit its viral replication and entry in vitro and in vivo [69,70]. HSV-1 infection was implicated as a disease risk factor foremost in AD but also in PD in different in vitro and in vivo investigations [71,72]. A 2.56-fold increased risk of developing dementia was reported in a retrospective cohort study with 8362 patients with acute HSV-1 or HSV-2 infections [60]. A phase 2 study investigating whether valaciclovir can slow the progression of AD in patients with HSV-1 is currently ongoing (clinicaltrials.gov NCT03282916) [70].

There are different studies suggesting an involvement of the adaptive immune system in the development of neurodegeneration. Genome-wide association studies have found an association of specific major histocompatibility complex II gene alleles with PD and T-cells of PD patients were shown to react to α-synuclein epitopes [73]. Another group showed that Th17-T-cells contribute to PD pathogenesis in a cell culture model of PD using induced pluripotent stem cells (iPSCs) [74]. Recently, T-cells were found to be adjacent to Lewy bodies and dopaminergic neurons in brains of Lewy-body-dementia patients and stimulation of CD4^+^ T-cells with a phosphorylated α-synuclein epitope resulted in increased IL-17 production as a sign of a Th17-response [75].

Take home messages of Chapter 2 (Section 4):Multiple epidemiological studies link different (viral) infections to PD, as individuals with certain infections have an elevated risk for PD.The protein α-synuclein might physiologically act as an infection defense mechanism, entrapping viral particles, which could lead to its pathological aggregation and subsequent neurotoxic PD effects.The involvement of the adaptive immune system in the development of neurodegenerative diseases has been increasingly implicated supporting the hypothesis that infections, and thus activation of the immune system can trigger neurodegenerative cascades.

## 5. Chapter 3

### 5.1. General Implications of SARS-CoV-2 in Neurodegeneration

The previously discussed mechanisms of viral neurotropism and neuroinflammation raise the question of whether long-term neurodegeneration has to be expected after COVID-19 disease.

SARS-CoV-2 and potentially pathogenic proteins involved in neurodegeneration have been linked by different studies. It was observed that the spike protein receptor binding domain binds to heparin and heparin binding proteins including amyloid-β, α-synuclein, tau, prion and TDP-43, which may initiate the pathological aggregation of these proteins resulting in neurodegeneration [76,77]. The same mechanism is described for HSV-1, which catalyzes the aggregation of amyloid-β in vitro and in vivo and is a well-established risk factor for AD [76,78]. Recently, it was demonstrated that viral particles (including SARS-CoV-2 spike protein) facilitate the spreading of proteopathic seeds by altering intercellular cargo transfer [79].

Viruses use different strategies to take control over host cellular functions, such as interfering with autophagy and mitochondrial or lysosomal functions, which are implicated in the development of neurodegenerative disease as well [80]. SARS-CoV-2 alters autophagy and mitochondrial and lysosomal functions in infected lung cells [81].

Furthermore, viral changes of proteostasis of the host cell can lead to accelerated “aging” of the infected tissue, which may then boost neurodegenerative processes that are common in senescent cells [80].

Ferrosenescence is an iron-mediated premature aging process of cells that results in an the iron-induced disruption of DNA repair and, thus, in neurodegeneration [82]. An interesting aspect of viral capabilities is the induction of ferrosenescence in host cells to facilitate viral replication [82].

These data support the notion that SARS-CoV-2 infections can induce alterations promoting neurodegenerative cascades.

### 5.2. COVID-19 and Possible Mechanisms Connected to Parkinson’s Disease

There are several links between COVID-19 and the development of PD that are elaborated in this section.

In 1985, it was observed that infection of mice with the mouse hepatitis virus (that has been identified as a murine analogue of the human Coronaviridae) resulted in mild encephalitis and the deposition of viral antigens mostly in the nucleus subthalamicus and the substantia nigra [83]. This led to subsequent gliosis in those regions, suggesting a link between the virus and PD/postencephalitic parkinsonism [83]. Antibodies against Coronaviridae were found to be elevated in the CSF of PD patients compared to controls as early as 1992 [84].

Thus far, three case reports of PD onset in timely correlation to COVID-19 disease have been reported; however, a clear causal link could not be established [85].

Two cases of patients developing COVID-19-associated encephalitis with progressive atypical parkinsonism and FDG-PET alterations reminiscent of postencephalitic parkinsonism were published recently [86].

Several mechanisms by which COVID-19 might contribute to the development of PD were previously reviewed and discussed: Vascular insults in the nigrostriatal system could lead to subsequent parkinsonism [87]. Furthermore, the cytokine storm associated with severe COVID-19 triggers neuroinflammation and, subsequently, neurodegeneration [33,87]. Systemic levels of IL-6 are elevated in COVID-19, and a small prospective observational study revealed that a higher level of IL-6 was associated with an increased risk of developing PD [88].

Another possible mechanism of inducing PD would be viral neurotropism resulting in direct neuronal damage in strategic areas. IPSC-derived midbrain dopaminergic neurons were shown to be submissive to SARS-CoV-2-infection, which triggered an inflammatory response and subsequently cellular senescence in vitro [89]. RNA-sequencing analysis of the ventral midbrain tissue of COVID-19 patients revealed a comparable phenotype of inflamed neurons and identified low levels of SARS-CoV-2 transcripts [89]. These data underline that there may be a special susceptibility to SARS-CoV-2 of particularly vulnerable midbrain regions involved in the development of PD.

The general susceptibility of central nervous structures to SARS-CoV-2 was shown by Ramani et al., who infected brain organoids and observed viral entry especially in neurons. The infection induced an altered distribution and hyperphosphorylation of tau and subsequent neuronal death [90].

A link between NF-κB and PD was previously established because NF-κB was increased in the substantia nigra of MPTP-treated mice [91]. MPTP treatment is a common animal model of PD as the neurotoxin leads to nigrostriatal degeneration and loss of dopaminergic neurons [91,92]. Suppressing NF-κB in this model led to prevention of the degeneration of dopaminergic neurons [91]. In an in vitro model of dopaminergic neurons, treatment with 6-OHDA led to NF-κB activation, caspase activation and apoptotic death that was prevented by inhibition of NF-κB [93]. NF-κB is activated by SARS-CoV-2 via pattern recognition receptors, which might be a neurodegenerative trigger [93].

Other interesting aspects are the shared implications for the angiotensin–aldosterone system in COVID-19 and PD. Angiotensinogen is produced by astrocytes as part of a local independent renin–angiotensin system (RAS) [94,95]. Its pathological overactivation (that also results from the degeneration of dopaminergic neurons) led to oxidative stress and inflammation, whereas its inhibition was considered as a treatment option in several neurodegenerative diseases including PD and AD [96,97]. SARS-CoV-2 uses the ACE2-receptor as an entryway into host cells and, therefore, intervenes with the RAS as well [10].

A previously observed connection between H1N1 influenza virus and α-synuclein aggregation could potentially be relevant for SARS-CoV-2, too. H1N1 led to the aggregation of endogenous α-synuclein in LUHMES cells [98]. As a reason for the pathological α-synuclein aggregation following H1N1 infection, an impairment of the autophagosome of infected LUHMES-cells was proposed [98]. Interestingly, α-synuclein aggregates were also seen in the olfactory bulb after intranasal instillation of H1N1 [98].

Early symptoms of PD are olfactory and vegetative dysfunction including obstipation as well as the prodromal syndrome REM sleep behavior disorder (RBD). Olfactory dysfunction is a very common early symptom of COVID-19, and the olfactory route is discussed as one way of viral entry into the CNS [21,26]. Therefore, it seems plausible that COVID-19 might influence the pathogenesis of PD as SARS-CoV-2 can take a route of spreading that was described for the developing neuropathology in PD [99,100].

Polysomnographic investigations in 11 patients four months after initial infection with SARS-CoV-2 revealed episodes of REM sleep without atonia in 4 patients, which is a characteristic (prodromal) sign of RBD [101].

Another interesting aspect is that the development of PD is linked to the gut microbiome and its dysbiosis [102]. SARS-CoV-2 causes an imbalance of the gut microbiome (dysbiosis) and intestinal inflammation indicated by elevated fecal calprotectin in COVID-19-associated diarrhea, which proposes a possible link to PD [103,104]. SARS-CoV-2 RNA was detected in the feces of about 50% of patients with COVID-19, supporting the hypothesis of intestinal infection [105].

Molecular investigations have established links between COVID-19 and PD focusing on protein interactions. In all, 44 proteins in the CNS implicated in PD were found to interact with 24 host proteins from the lung that interact with SARS-CoV-2 viral proteins [106]. The two most promising interaction candidates were Rab7a and NUP62 [106]. Rab7a is a lysosomal protein reducing the proportion of cells with α-synuclein particles as well as the toxicity of α-synuclein, whereas NUP62 is involved in autophagosome formation [106]. The comparison of transcriptomic modulations induced by SARS-CoV-2 and PD also revealed significant overlap in several pathways [107].

On the other hand, a protective role of α-synuclein against COVID-19 was proposed since α-synuclein, like β-amyloid, is upregulated in the face of viral infections and can restrict viral replication acting as a defense mechanism in the brain [108]. This leads to the speculation that PD patients with higher α-synuclein levels in the brain might have some protection against SARS-CoV-2 infection [109]. Before the COVID-19 pandemic, a Japanese retrospective cohort study showed that hospitalized PD patients were less likely than other patients to die from pneumonia [109].

If the viral infection leads to an upregulation of α-synuclein as a defense mechanism, it may induce prolonged inflammation and neuronal death triggering the development of PD on the long run as it was shown earlier for West Nile virus infections [87].

Interestingly, a hypothetical connection between COVID-19 and atypical parkinsonism can be established as well, although data on this topic are rare so far. It was demonstrated that atypical Parkinson syndromes such as multisystem atrophy and progressive supranuclear palsy are associated with microglial activation as a sign of neuroinflammation and that the microglial activation contributes to the progression of neurodegeneration [110,111,112]. Recently, it was shown that microglial activation can be visualized by PET imaging, which might function as a biomarker for tauopathies [113,114]. Microglial activation and neuroinflammation are seen in COVID-19 as described in chapter 1, creating a link between atypical parkinsonism and COVID-19 [39].

### 5.3. Alzheimer’s Disease, Cognitive Deficits and COVID-19

There is cumulating evidence demonstrating a close connection between cognitive disturbances and COVID-19. A prospective longitudinal study revealed that cognitive decline measured by Montreal Cognitive Assessment (MOCA) was apparent in 21% of mild COVID-19 patients vs. 2% of seronegative individuals [115]. Another study found pathological MOCA results in 18 of 26 COVID-19 patients and also FDG-PET abnormalities (frontoparietal hypometabolism) in 10 patients matching the clinical deficit [45].

Cognitive decline was not only observed during acute infection, but there are also reports of persistent cognitive impairment after recovering from COVID-19, as MOCA abnormalities were detected in a group of post-COVID-19 patients [116]. Another study confirmed that cognitive deficits persisted in 70% of COVID-19 patients for at least 1 month after hospital discharge [116,117]. Similarly, 46 of 57 recovering COVID-19 patients (81%) had signs of cognitive impairment [13,118]. Interestingly, persisting memory and concentration deficits were found after SARS-Cov-1 and MERS infections in 15–20% of cases [119].

Another interesting study used transcranial magnetic stimulation to investigate recovered COVID-19 patients who suffered from severe disease with ICU stay and neurological complications reporting fatigue and showing abnormal scores in the frontal assessment battery during the subacute phase [120]. The transcranial magnetic stimulation in these patients revealed severe impairment of GABAergic intracortical circuits while glutamatergic transmission was intact [120]. GABAergic impairments are usually common in frontotemporal dementia and executive dysfunction [120,121]. However, it has to be noted that cognitive impairment is a common problem after suffering from acute respiratory distress syndrome (ARDS), which can have multiple reasons other than COVID-19 [116,122,123,124]. After ARDS, cognitive disturbances persisted in long-term follow-ups in about 10% of cases [116,122]. Other studies found cognitive deficits and psychiatric disorders (mainly depression and anxiety) in up to 60% of ARDS survivors after 12 months [125].

Dementia was found to be one of the strongest risk factors for COVID-19 and associated with a higher mortality [126,127,128,129,130]. Apparently, patients with dementia have difficulties in following hygiene rules, mask requirements, behavioral instructions and distancing rules due to cognitive deficits [124,131]. Dementia patients frequently live in nursing homes where a higher risk for infection with the virus was present in many areas [124]. COVID-19 disease in dementia patients often appeared atypical presenting foremost with delirium/confusion and few infectious symptoms [129,132]. Confusion and mood and behavioral disturbances persisted in 19.2% of survivors [129].

An analysis of the network-based relationship for the gene/protein sets between virus and host factors as well as different neurological diseases in an interactome network model showed close proximity between COVID-19 and cognitive decline as well as AD and PD [13].

Postmortem studies demonstrated that ACE2-expression was increased in the brains of AD patients. Especially in severe dementia, ACE2-expression was elevated, which could lead to a higher susceptibility for COVID-19 [123,124,133].

Ischemic white matter damage occurs early in AD contributing to the progression of dementia. COVID-19 can induce vascular lesions due to hypercoagulability and can be expected to accelerate disease progression in AD patients [123,134].

It was hypothesized that amyloid-β, the protein implicated in AD development, is an antimicrobial peptide involved in fighting cerebral SARS-CoV-2 infection as previously described for α-synuclein in PD [123,135]. It could be speculated that amyloid-β is upregulated as a defense mechanism during infection leading to an overactivation with pathological deposition of amyloid-β in the long run [123,135].

Apoε4, an established risk factor for AD, was also recognized as a prominent risk factor for COVID-19, potentially linking the two pathophysiologies [136]. COVID-19 severity could be statistically predicted by the Apoε4 genotype [136]. In human iPSCs models with Apoε4 genotype, neurons and astrocytes were more susceptible to SARS-CoV-2 infection than non-Apoε4 cells and brain organoids [12].

Another point of overlap is IL-6, which was shown to be elevated in COVID-19 and was also considered a biomarker with prognostic value in AD [49,123,137].

SARS-CoV-2 is likely to disturb autonomic functions in vagal regulation centers in the brainstem [127]. In AD, autonomic functions are impaired as well, since higher cardiac sympathetic function and lower parasympathetic function are reported in patients [127]. Therefore, noninvasive (auricular) vagal nerve stimulation is discussed as a therapeutic strategy for AD as well as severe COVID-19, since a downregulation of inflammatory pathways (reduction of IL-6 levels) is expected as a result [127]. Supporting this theory, transauricular vagal nerve stimulation was able to reduce cognitive dysfunction in a preclinical murine model of AD [138].

AD leads to alterations in calcium homeostasis in the brain; RNA-viruses use the same mechanism to facilitate viral replication [124]. Therefore, viral replication might be easier in the brain of AD patients where calcium homeostasis is already abnormal [124].

There is an association between AD and diabetes type II, which elevates the risk to develop AD [139]. AD and diabetes type II were both found to be strong risk factors for COVID-19 endangering those patient groups particularly and proposing a mechanistic link between these diseases that could explain an overlapping pathophysiology [139].

Take home messages of Chapter 3 (Section 5):Viruses have different strategies to take control over host cellular functions, for example, impairing autophagy and mitochondrial or lysosomal properties, whose dysfunction has been implicated in neurodegenerative diseases.Neuroinflammatory alterations due to COVID-19 such as elevated IL-6 levels or activation of NF-κB might trigger/accelerate the development of PD.Direct CNS invasion by SARS-CoV-2 might also lead to induction of neurodegenerative cascades in strategic areas.COVID-19 can lead to acute and persisting cognitive deficits, although some of those might be due to ARDS.Dementia and Apoε4 genotype are strong risk factors for COVID-19 and its associated mortality.

## 6. Chapter 4

### Parkinson’s Disease and COVID-19: Effects on PD Symptoms, Psychological and Social Aspects

The impact of the pandemic on aspects of daily life was massive for the entire world population, and patients with chronic diseases in need for regular care were especially affected. An extensive analysis of worldwide studies (210,419 participants total) showed that acute care for neurological disorders in general was disrupted due to the pandemic in 47.1% of cases [140]. The impact on PD patients was differentially described, as specific problems occurred in relation to pandemic-associated restrictions. Psychological issues as well as aspects regarding care and supply of medication were found to be most burdensome in this cohort [140].

COVID-19 has the ability to alter the pharmacodynamics of levodopa also due to diarrhea, which is a common symptom of COVID-19 [141]. This leads to motor fluctuations in infected PD patients [141]. PD patients suffering from COVID-19 often developed a post-COVID syndrome (85.2%) consisting of worsened motor functions, increased daily levodopa dose requirement, fatigue, loss of concentration and sleep disturbances [142].

However, subjective worsening of motor and non-motor symptoms of uninfected PD patients during the time of the pandemic was also recorded in different studies [143,144]. New behavioral symptoms were observed in 26% of PD patients in an Italian cross-sectional study [144]. PD patients reported feeling lonely and deprived of the support and communication with their physician [143].

It was hypothesized that dopamine-dependent adaptation is a requirement for successful coping; thus, PD patients are cognitively less flexible and can have more difficulties to adapt to new environments [145,146]. Therefore, the pandemic may lead to a relevant amount of stress in PD patients who are forced to adapt to a new environment quickly. Psychological stress was shown to worsen PD symptoms as well as the efficacy of dopaminergic medication especially on the tremor [145,147]. This could be an explanation for symptom exacerbation in PD patients during the pandemic.

It was found that 103 PD patients reported four main problems in the first Italian lockdown: 1. fear of contracting corona, 2. reduction of physical activity, 3. not being able to access support services and clinics and 4. reduction of socialization [148]. There was an objective reduction of physical activity, measured by a smartphone application, as most PD patients failed to meet 30 min of activity per day [149]. This was aggravated further in 44% during confinement [149]. It is well known that physical activity and training is an important treatment strategy in PD to maintain motor functions and independence, so the deprivation of physical activity during a lockdown can be suspected to lead to symptom progression and loss of independence [149].

Moreover, 66% of PD patients in a large cohort at Columbia University reported mood and sleep disturbances in the face of the pandemic; depression and insomnia were the most frequently reported psychiatric symptoms in multiple other studies as well [150,151,152,153,154]. A Chinese study revealed that PD patients had more sleep disturbances and anxiety than healthy controls and that these symptoms were independently associated with an exacerbation of other PD symptoms [154]. Sleep problems were also associated with a poorer quality of life [153].

Mindfulness-based interventions were shown to reduce depression and anxiety, improve motor function and strengthen resilience [145]. As this can be accomplished virtually, it appears to be a useful treatment strategy now and for the future [145].

The hours of caregiving increased dramatically during the pandemic. Care was mostly provided by family members [155,156]. Caregiver burden was increased during the COVID-19 era [144]. Interestingly, Montanaro et al. and others showed that anxiety and depression were frequent in both PD patients and their caregivers [157,158]. Depression was observed in 35% of PD patients and 21.7% of caregivers; 39% of PD patients and 40% of caregivers suffered from anxiety [158]. Therefore, caregivers should receive more support particularly during this pandemic to cope with their own burden and the neuropsychiatric symptoms of their relatives [159].

However, COVID-19 does not only have an impact on the PD symptoms; it was also discussed that pre-existing PD can elevate the risk of mortality or case fatality when an infection with SARS-CoV-2 occurs. The data on this topic are controversial (subsumed in Table 1).

A detailed review by Fearon et al. summed up that COVID-19 mortality is probably not increased in PD patients who tended to experience less dyspnea during the infection, were completely asymptomatic more often and less likely to require hospitalization [141,160,165,166,167]. The duration of ICU stays/hospitalization and ventilation did also not differ in PD and non-PD COVID-19 patients in a large analysis of German inpatients [165]. An Italian study compared COVID-19 patients with PD to COVID-19 patients without PD and found no difference in mortality (5.7% of PD COVID-19 patients died vs. 7.6% of non-PD COVID-19 patients) [160].

This trend could be supported by the hypothesis that amantadine and entacapone might have protective value against COVID-19, which was suggested by different studies [172,173,174]. However, a systematic review about a total of 1061 PD patients with confirmed COVID-19 showed a higher hospitalization rate, case fatality and mortality for these patients than that for non-PD COVID-19 patients [175]. A limitation of this study was the missing age matching, which is likely to influence the result, as age is one of the most established risk factors for case fatality and mortality of COVID-19 [175,176]. An American study compared 78,355 non-PD COVID-19 patients to 694 COVID-19 patients with PD and found an increased mortality even after adjusting and matching to age and sex [169]. A multicentric German study showed that prevalence and mortality of COVID-19 was higher in PD than in non-PD inpatients [168].

Apparently, these data are inconclusive and a definite suggestion on whether PD patients are more at risk for a (severe) COVID-19 infection cannot be made at this point. It should be noted that PD patients who suffer from COVID-19 infection are likely to present with atypical symptoms such as mood changes, fatigue, joint pain, flushing and exacerbation of PD-symptoms, which can complicate the diagnosis of SARS-CoV-2 infection [177].

Take home messages of Chapter 4 (Section 6):Post-COVID-syndrome, altered pharmacodynamics of levodopa and worsening of motor-symptoms are common in PD patients with COVID-19.Even uninfected PD patients often suffered from subjective worsening of motor and non-motor symptoms, reduced physical activity, as well as increased stress, anxiety and depression.Whether PD elevates the risk of COVID-19 mortality is not clear yet, since data on this topic is inconclusive.

## 7. Concluding Remarks

The overview presented in this review underlines that COVID-19 has a cerebral/neurological impact. Whether SARS-CoV-2 can enter the CNS and impose direct neuronal damage or whether neurological symptoms are rather due to secondary effects cannot be differentiated with certainty. The influence of the infection on neurodegenerative diseases is also not clear yet, but various pathophysiological theories exist linking COVID-19 to neurodegeneration and making it appear likely that the pandemic can have an (accelerating) influence on neurodegenerative diseases such as AD and PD. The same was proposed for other viral infections in the past, though a clear triggering/accelerating influence of viral infections on neurodegeneration could only be demonstrated for few viruses such as HCV and HIV so far. Monitoring recovered COVID-19 patients and especially patients with neurodegenerative diseases and COVID-19 will hopefully answer some of these questions in the future. Such a prospective investigation also withholds the potential to learn more about the neurodegenerative pathophysiology and develop new strategies for disease-modifying treatments.

## Figures and Tables

**Table 1 brainsci-11-01654-t001:** Mortality and morbidity in COVID-19 PD cases versus COVID-19 non-PD cases.

Authors	Country	Type of Study	No. of Patients	Results
Fasano et al., 2020 [160]	Italy	Case control	n = 105 PD + COVID, n = 92 controls	- PD/non-PD with no difference in COVID-19 prevalence (7.1% vs. 7.6%) and mortality (5.7% vs. 7.6%) - COVID-19 symptoms in PD milder compared to non-PD
Del Prete et al., 2020 [161]	Italy	Case-controlled survey	n = 740 PD cases, 7 infected with COVID	- Mortality 14% in COVID-infected PD-cases (n = 1) - Age-stratified mortality in line with national data (14%)
Fasano et al., 2020 [162]	Italy, Iran, Spain, UK	Multicenter cohort	n = 117 PD + COVID	- Mortality rate 19.4% (PD + COVID)
Antonini et al., 2020 [163]	Italy	Case series	n = 10 PD + COVID	- Mortality rate 40% (n = 4) - Higher mortality when older/longer PD-duration/DBS/levodopa infusion therapy
Cilia et al., 2020 [164]	Italy	Case control	n = 12 PD + COVID, n = 36 controls	- Mortality 0% - Higher PD-associated morbidity
Huber et al., 2021 [165]	Germany	Prospective multicenter cohort study	n = 40 PD + COVID, n = 600 matched controls	- COVID-19-associated mortality (32.5% PD vs. 26.7% non-PD) and morbidity not significantly different
Buccafusca et al., 2021 [166]	Italy	Case series	n = 12 PD + COVID	- Mortality 0%
Vignatelli et al., 2021 [167]	Italy	Case control	n = 696 PD + COVID, n = 8590 controls	- 30 days case fatality rate comparable (PD 35.1% vs. controls 39%)
Scherbaum et al., 2021 [168]	Germany	Cross-sectional study	n = 693 PD + COVID, control number unknown	- COVID-19 mortality rate: PD 35.4% vs. 20.7% controls (*p* < 0.001)
Zhang et al., 2020 [169]	USA	Case control	n = 694 PD + COVID, n = 78,355 controls	- Mortality rate 21.3% PD vs. 5.5% non-PD (*p* < 0.001) - 1:5 matching: PD patients with significantly higher mortality from COVID
Zhai et al., 2020 [170]	China	Single-center retrospective study	n = 10 PD + COVID, n = 286 controls	- Inclusion of severe or critical ill COVID patients - Mortality rate PD 30% vs. non-PD 40.2% (*p* > 0.05)
Artusi et al., 2020 [171]	Italy	Case series	n = 8 PD + COVID, control “general population”	- Higher mortality (75% PD vs. 12% controls) - PD symptoms worsened in all patients

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
