# Peer review of "Can SARS-CoV-2 Infection Lead to Neurodegeneration and Parkinson’s Disease?"

_brainsci, 2021, doi:10.3390/brainsci11121654_

Round 1

Reviewer 1 Report

This manuscript summarizes available evidence on the relation between COVID-19 and Parkinson’s disease (PD), with a focus on four topics: putative effects of (i) SARS-COV-2 or (ii) other viruses that may induce neurodegeneration; (iii) associations between COVID-19 and clinical neurodegenerative diseases, including PD ; and (iv) the psychosocial impact of COVID-19 on people with PD.

The topic of this review is timely, given the burden that COVID-19 continues to pose on societies across the world, in particular on people with -or prone to- neurodegenerative diseases. However, the scope of the manuscript is very broad and lacks an in-depth qualitative analysis of either of the four topics which are at the core of this review. Furthermore, it is unclear which selection process was used to identify suitable studies for this review (e.g., systematic search in PubMed) and how the methodological quality of the the selected studies was assessed. 

To address these issues, I strongly suggest that the authors provide (in the revised manuscript) an overview of takeaway messages per topic, a description of the article selection process and a description of the assessment tool they used to appraise the quality of the available studies.

Author Response

Response to Reviewer 1

To address these issues, I strongly suggest that the authors provide (in the revised manuscript) an overview of takeaway messages per topic, a description of the article selection process and a description of the assessment tool they used to appraise the quality of the available studies.

We thank the reviewer for these helpful suggestions and agree that take-away-messages of every chapter make our manuscript clearer and give additional focus that will be helpful to the reader.

We included the following take-home-messages concerning each chapter:

Take-home-messages of Chapter 1:

  1. Since it was shown for other human coronaviruses in the past, it is likely that SARS-Cov-2 can be neurotropic.
  2. There are three possible routes of neuroinvasion: The transneural route via the olfactory nerve, the hematogenous route via vascular endothelium or a permeable blood-brain-barrier and the “Trojan-horse-mechanism” via infiltration of immune cells and subsequent invasion into the CNS via diapedesis.
  3. Neuropathologically, SARS-Cov-2 leads to microglial activation in distinct areas.
  4. SARS-Cov-2 triggers a neuroinflammatory response with elevated serum levels of several cytokines (e.g. IL1 and IL-6) and elevated markers indicating CNS lesions like NfL and GFAP in CSF.

Take-home-messages of Chapter 2:

  1. Multiple epidemiological studies link different (viral) infections to PD, as individuals with certain infections have an elevated risk for PD.
  2. α-Synuclein might physiologically act as a viral defense mechanism entrapping viral particles, this might however lead to its pathological aggregation and subsequent neurotoxic effects.
  3. The involvement of the adaptive immune system in the development of neurodegenerative diseases is implicated increasingly; this supports the hypothesis that infections and thus activation of the immune system can trigger neurodegenerative cascades.

 Take-home-messages of Chapter 3:

  1. Viruses have different strategies to take control over host cellular functions like impairing autophagy, mitochondrial or lysosomal function whose dysfunction is implicated in neurodegenerative diseases.
  2. Neuroinflammatory alterations due to COVID-19 like elevated IL-6 levels or activation of NFκB might trigger/ accelerate the development of PD.
  3. Direct CNS invasion by SARS-Cov-2 might also lead to induction of neurodegenerative cascades in strategic areas.
  4. COVID-19 leads to acute and persisting cognitive deficits, though some of those might be due to ARDS.
  5. Dementia and Apoε4 genotype are strong risk factors for COVID-19 and COVID-19 associated mortality.

 Take-home-messages of Chapter 4:

  1. Post-COVID-syndrome, altered pharmacodynamics of levodopa and worsening of motor-symptoms are very common in PD patients with COVID-19.
  2. Even uninfected PD patients often suffered from subjective worsening of motor- and non-motor symptoms, reduced physical activity, as well as increased stress, anxiety and depression.
  3. Whether PD elevates the risk of COVID-19 mortality and case fatality is not clear yet, since data on this topic is inconclusive.

Furthermore we added a “methods”-paragraph describing the selection process of the articles as you can see below. Since this article was not designed as a systematic review, we did not use specific guidelines/ tools to extract articles and check their quality. We only included articles published in English in international peer-review journals and evaluated the articles by content/ topic and impact factor individually.

Now it reads in line 38 – 45:

2. Methods

Literature research for this review was done in Pubmed using the search terms “COVID-19”, “SARS-Cov-2”, “Parkinson’s disease”, “Alzheimer’s disease”, “neurodegeneration”, “Viral infection” and “ infection” in different combinations. Only articles published in English in international peer-review journals were included into the selection process. Articles were selected by screening the abstracts for eligibility to the topic; articles contributing interesting information to the core topics of this review were included.

Reviewer 2 Report

The Authors present a review considering the impact of COVID-19 on PD, AD and neurodegeneration. Paper includes i.a. description of possible mechanisms of neuroinfection and neuropathologic changes observed in post-mortem studies. Authors included section describing the role of SARS-CoV2 infection in dementia and AD. Paper is well-written. However, I believe the manuscript would benefit, if a paragraph concerning atypical parkinsonian syndromes was added. The possible role of neuroinflammation in pathogenesis of APS is currently widely discussed in the literature, and the impact of COVID infection in this area would be important argument in the discussion. What is more, due to difficulties in ante-mortem differential diagnosis, it is common that patients with APS are misdiagnosed with PD, especially in early stages of the disease.  
I would recommend taking into consideration papers listed below:

Alster P, Madetko N, Koziorowski D, Friedman A. Microglial Activation and Inflammation as a Factor in the Pathogenesis of Progressive Supranuclear Palsy (PSP). Front Neurosci. 2020 Sep 2;14:893. doi: 10.3389/fnins.2020.00893. PMID: 32982676; PMCID: PMC7492584.

Malpetti M, Passamonti L, Jones PS, Street D, Rittman T, Fryer TD, Hong YT, Vàsquez Rodriguez P, Bevan-Jones WR, Aigbirhio FI, O'Brien JT, Rowe JB. Neuroinflammation predicts disease progression in progressive supranuclear palsy. J Neurol Neurosurg Psychiatry. 2021 Jul;92(7):769-775. doi: 10.1136/jnnp-2020-325549. Epub 2021 Mar 17. PMID: 33731439; PMCID: PMC7611006.

Kübler D, Wächter T, Cabanel N, Su Z, Turkheimer FE, Dodel R, Brooks DJ, Oertel WH, Gerhard A. Widespread microglial activation in multiple system atrophy. Mov Disord. 2019 Apr;34(4):564-568. doi: 10.1002/mds.27620. Epub 2019 Feb 6. PMID: 30726574; PMCID: PMC6659386.

Author Response

Response to Reviewer 2

However, I believe the manuscript would benefit, if a paragraph concerning atypical parkinsonian syndromes was added. The possible role of neuroinflammation in pathogenesis of APS is currently widely discussed in the literature, and the impact of COVID infection in this area would be important argument in the discussion. What is more, due to difficulties in ante-mortem differential diagnosis, it is common that patients with APS are misdiagnosed with PD, especially in early stages of the disease.

I would recommend taking into consideration papers listed below:

Alster P, Madetko N, Koziorowski D, Friedman A. Microglial Activation and Inflammation as a Factor in the Pathogenesis of Progressive Supranuclear Palsy (PSP). Front Neurosci. 2020 Sep 2;14:893. doi: 10.3389/fnins.2020.00893. PMID: 32982676; PMCID: PMC7492584.

Malpetti M, Passamonti L, Jones PS, Street D, Rittman T, Fryer TD, Hong YT, Vàsquez Rodriguez P, Bevan-Jones WR, Aigbirhio FI, O'Brien JT, Rowe JB. Neuroinflammation predicts disease progression in progressive supranuclear palsy. J Neurol Neurosurg Psychiatry. 2021 Jul;92(7):769-775. doi: 10.1136/jnnp-2020-325549. Epub 2021 Mar 17. PMID: 33731439; PMCID: PMC7611006.

Kübler D, Wächter T, Cabanel N, Su Z, Turkheimer FE, Dodel R, Brooks DJ, Oertel WH, Gerhard A. Widespread microglial activation in multiple system atrophy. Mov Disord. 2019 Apr;34(4):564-568. doi: 10.1002/mds.27620. Epub 2019 Feb 6. PMID: 30726574; PMCID: PMC6659386.

We thank the reviewer for this interesting suggestion and included a paragraph regarding this aspect in chapter 3 as you can see below. There is a hypothetical connection between COVID-19 and atypical parkinsonism as neuroinflammation and especially microglial activation was implicated in APS and was also shown to occur in COVID-19.

Now it reads in chapter 3, page 8 (lines 376 - 384): Interestingly, a hypothetical connection between COVID-19 and atypical parkinsonism can be established as well, although data on this topic are rare so far. It was demonstrated that atypical Parkinson syndromes like multisystem atrophy and progressive supranuclear palsy are associated with microglial activation as a sign of neuroinflammation and that the microglial activation contributes to the progression of neurodegeneration [110–112]. Recently, it was shown that microglial activation can be visualized by PET-imaging which might function as a biomarker for tauopathies [113,114]. Microglial activation and neuroinflammation is seen in COVID-19 as described in chapter 1, creating a link between atypical parkinsonism and COVID-19 [39].

Round 2

Reviewer 1 Report

The authors have adequately addressed my concerns

Reviewer 2 Report

I endorse publication of this manuscript.